# The Phylogenetic Relationships of the Fanniidae within the Muscoid Grade (Diptera: Calyptrata) Based on the Musculature of the Male Terminalia

**DOI:** 10.3390/insects13020210

**Published:** 2022-02-18

**Authors:** Vera S. Sorokina, Olga G. Ovtshinnikova

**Affiliations:** 1Institute of Systematics and Ecology of Animals, Siberian Branch of the Russian Academy of Sciences, 630091 Novosibirsk, Russia; 2Department of Invertebrate Zoology, Biological Institute, Tomsk State University, 634050 Tomsk, Russia; 3Zoological Institute, Russian Academy of Sciences, 199034 St. Petersburg, Russia; brach@zin.ru

**Keywords:** Calyptrata, muscoid grade, flies, male genitalia, muscles, pregenital segments, sclerites

## Abstract

**Simple Summary:**

The Fanniidae is a small dipteran family in the muscoid grade of the Calyptrata. To resolve controversial issues of phylogeny, in addition to molecular analyses, the study of the muscles of the pregenital and genital segments of males can significantly help because they are considered to provide important information relating to the phylogenetic branching of the Cyclorrhapha. Here, the authors report on the structure of the sclerites and the muscles of terminal segments of three species of the Fanniidae. These structures are compared with other members of the muscoid grade, as well as the Oestroidea and Hippoboscoidea. In comparison with the majority of the Calyptrata, there is a significant reduction of the sclerites and muscles of the pregenital segments of the male genitalia of the Fanniidae. The presence of the lateral bacilliform sclerites, as well as the presence and position of the epandrial muscles of the Fanniidae as in primitive Muscidae and also in *Tachina* and *Calliphora*, is best interpreted as a plesiomorphic state, while the reductions are considered as derived relative to the ground plan of the Calyptrata.

**Abstract:**

The abdominal and pregenital segments and the genitalia were studied in males of *Fannia subpellucens* (Zetterstedt, 1845), *Fannia canicularis* (Linnaeus, 1761) and *Fannia incisurata* (Zetterstedt, 1838). In comparison with the remaining members of the muscoid grade, in addition to the symmetry of the pregenital segments, significant reductions of the sclerites and musculature of the male terminalia have been observed in Fanniidae. The muscular structure of pregenital segments confirms that the fused pregenital ring is syntergosternite VI + VII + VIII. Symmetry and fusion, as well as the lower number of the sclerites and muscles of the pregenital segments and male genitalia of the Fanniidae, can be considered apomorphic character states. The presence of the lateral bacilliform sclerite, as well as the presence and position of the epandrial muscles M 26, three pairs of muscles M 19 and paired muscles M 18, can be considered as a plesiomorphic character state of the Fanniidae. The structure of the sclerites and muscles of the male abdominal segments and terminalia place the Fanniidae at the base of the muscoid grade and Oestroidea, as has been confirmed by recent molecular studies.

## 1. Introduction

The Fanniidae is a small family of the Calyptrata in the paraphyletic assemblage of families here referred to as the muscoid grade. The world fauna includes some 410 species in five genera that are distributed across all zoogeographic regions, but the greatest numbers of species are found in the Holarctic and Neotropical regions [1,2,3]. Adult Fanniidae can be found in various environments, but they mostly prefer forests where the males often hover outdoors, either singly or forming swarms near the lower branches of trees and above forest paths. Females are found in the ground vegetation and can be collected by sweeping. Fanniidae can be attracted to aphid honeydew, plant sap, various flowers where they feed on nectar and pollen and light traps [1,4,5,6]. Females are usually attracted to decaying material, carrion and excrement, as well as sweat and mucus on animals and on humans, buzzing annoyingly around ears and eyes [1,4,6]. Some species of Fanniidae are synanthropic.

The larval habitats are very diverse. They include various decaying materials in gardens, domestic garbage, various fungi, rot holes in trees, decaying leaves and litter in forests, vertebrate and invertebrate carrion, cesspools, latrines and dung hills, human feces, animal droppings and dung, birds nests, burrows of vertebrates and nests of Hymenoptera, sandy and humus soil and animal tissue, causing myiasis [4,6]. As a result of the lifestyle of both larvae and adults of Fanniidae, they have considerable medical, hygienic, economic and forensic importance [1,6,7,8,9,10].

The systematic position of the Fanniidae within the Calyptrata and the muscoid grade has changed over the decades. In the 19th century, the Fanniidae were considered as the subfamily Homalomyinae of the Anthomyidae [sic] [11] and were then raised to family rank Homalomyidae by Schnabl and Dziedzicki [12], with subfamilies Azelinae and Fanninae. However, the latter proposal was not accepted, and several authors classified the taxon as a subfamily Fanniinae within the Muscidae [4,13,14,15]. Others recognized it as a distinct family within muscoids [1,6,16,17,18,19,20,21] and this rank of Fanniidae is currently accepted by most specialists.

The monophyly and the family rank of the Fanniidae have been confirmed based primarily on the morphological characters of the adults, as well as of the larvae [2,17,22,23,24] and then corroborated by DNA studies [25,26,27]. However, the phylogenetic relationships within the Fanniidae, as well as the systematic position of this family within the muscoid grade, have not been completely established. The intra-familial relationships of the Fanniidae based on the morphological similarities between-species groups were first analyzed by Chillcott [4], and cladistic analyses using adult external morphological characters and male terminalia were made by Hennig [22] and Domínguez and Roig-Juñent [2,28].

As regards the phylogenetic position of the Fanniidae within the muscoids, more traditional views placed the family as the sister group of the Muscidae [22,23,24,29]. Pont [1] suggested that the Fanniidae are the most plesiomorphic lineage of the muscoids and the sister group of the Scathophagidae + Anthomyiidae + Muscidae. Based on molecular data, the monophyletic Fanniidae are considered the sister group of the remaining non-hippoboscoid calyptrates ((Muscidae + (Anthomyiidae + Scathophagidae)) + Oestroidea) and are placed at the base of the muscoid grade [25,26,27,30,31]. The phylogenetic relationships within the muscoid grade thus remain controversial.

To resolve controversial issues of phylogeny, the study of the muscles of the pregenital and genital segments of males can significantly help because the male terminal skeleton and muscle transformations are considered to be of importance for the phylogenetic branching of the Cyclorrhapha [32,33,34]. In the Cyclorrhapha, both the sclerites and the muscles of pregenital segments VI–VIII and part of IX are asymmetrical as a result of the 360° clockwise rotation of the male genitalia. In this sense, the structure of the sclerites of the abdominal, pregenital and genital segments differs significantly among the various families.

Currently, the musculature of the male terminalia has only been studied in a few members of the muscoid grade, in particular in two species of Anthomyiidae [35], one species of Scathophagidae [33] and 16 species of Muscidae [32,36,37,38,39]. Although only a few members of this superfamily have been studied, tendencies in the reduction of the pregenital segments and musculature, as well as of the phallapodeme muscles, have been revealed in the evolution of this grade. A complete set of phallapodeme muscles in the Anthomyiidae and Scathophagidae corresponds to the plesiomorphic state, and therefore, the structure of the genital sclerites and muscles in the Muscidae shows a certain degree of reduction [39]. The homologies of the pregenital sclerites in members of different subfamilies of the Muscidae, in particular the nature of tergite VI, sternites VI and VII, syntergosternite VII + VIII and of the hypandrial appendages, were confirmed by analysis of the muscle connections. A different set of muscles has been detected within the Muscidae at the subfamily level [36,37,38,39].

The musculature of the male terminalia of the Fanniidae in relation to the other muscoids has not been previously studied. Such data could clarify the phylogenetic relationships within this superfamily. The general structure of the sclerites of the male terminalia of Fanniidae has been described in detail by many authors, e.g., [1,2,3,4,6,13,14,17,18]. However, the understanding of the presence of some structures, especially the sclerites of the aedeagal complex, and their terminology in these works is different. Primarily this concerns the male ejaculatory apodeme, epiphallus, basiphallus and pregonite, which are absent in this family according to most researchers, and the simpler structure of the male terminalia has been seen as a primitive state, e.g., [1,4,6]. Domínguez and Roig-Juñent [2], in their cladistic analyses of the Fanniidae, indicated that among the characters of the male terminalia which they used, the ejaculatory apodeme [18], epiphallus, basiphallus and pregonite were present in some species, but the authors did not indicate these structures for most of the relevant species. Unfortunately, there are no clear illustrations of the aedeagal complex with its associated structures in all the papers listed above, and this makes it difficult to understand correctly what is the true general structure of the sclerites of the male terminalia of the Fanniidae. Study of muscles of the pregenital, as well as of the genital, segments of males can significantly help resolve this issue.

This paper presents the results of our first study of the sclerites and muscles of the male abdominal segments and terminalia in members of the family Fanniidae: *Fannia subpellucens* (Zetterstedt, 1845), *F. canicularis* (Linnaeus, 1761), *F. incisurata* (Zetterstedt, 1838).

## 2. Materials and Methods

The material used in this paper is deposited in the collections of the Institute of Systematics and Ecology of Animals, Russian Academy of Sciences, Siberian Branch, Novosibirsk, Russia (SZMN) and the Zoological Institute, Russian Academy of Sciences, St. Petersburg, Russia (ZISP).

To study the genital sclerites, dry specimens were softened in a hydration chamber; the abdomen was then detached, treated with 10% KOH solution and dissected. The sclerites are designated here following the terminology of Sinclair [40] and Cumming and Wood [41].

The muscles of the male terminalia were studied by manual dissection of specimens preserved in 70% ethanol, using microknives, under a Leica MZ9.5 stereomicroscope (Wetzlar, Germany). The illustrations were made in Photoshop CS6 (Adobe Inc., San Jose, CA, USA), based on digital images of muscles and sclerites captured with a Canon EOS 77D (Tokyo, Japan) camera mounted on the Leica MZ9.5 trinocular head.

The muscles are classified into the following groups: abdominal, pregenital and genital muscles (tergosternal muscles, muscles of the hypandrial complex, and muscles of the epandrial complex). The muscles are numbered according to the classification of Ovtshinnikova [32,42] and grouped by the sites of their origin.

The following abbreviations are used in the text: bac scl—bacilliform sclerite; c—cercus; ej—ejaculatory apodeme; ep—epandrium; hyp—hypandrium; l—left muscle; r—right muscle; pgt—postgonite; phap—phallapodeme; prgt—pregonite; sbeps—subepandrial sclerite; sp—spiracle; st—sternite; stgst—syntergosternite; sur—surstylus; ISM—abdominal and pregenital intersegmental sternal muscles; ITM—abdominal and pregenital intersegmental tergal muscles; M1–M26—pregenital and genital muscles.

Because of genital rotation, the sclerites of the pregenital segments do not always lie in the usual position. For this reason, characteristics such as ‘wide’ or ‘narrow’ in the descriptions refer only to the geometric shape of the sclerites, regardless of their orientation relative to the body axis.

## 3. Results

The sclerites and musculature of the male terminalia of *Fannia subpellucens* (Zetterstedt, 1845), *F. canicularis* (Linnaeus, 1761) and *F. incisurata* (Zetterstedt, 1838).

### 3.1. Fannia Subpellucens (Zetterstedt, 1845)

#### 3.1.1. Material Examined

Six males, Russia, Magadan region, Koni Peninsula, environs of Cape Ploskyi cordon, 59°09′ N 151°38′ E, 1–6.vii.2017, leg. V. Sorokina.

#### 3.1.2. Description

*Abdominal segments.* Sternite I reduced to narrow band, tergites I and II fused. Segments III and IV and tergite V not modified; sternite V represented by two narrow longitudinal plates and membrane around these plates (Figure 1), connected to syntergosternite VI + VII + VIII.

*Pregenital segments* (Figure 1). Symmetrical. Tergite VI fused with syntergosternite VII + VIII; sternite VI fused with sternite VII forming narrow plate in boomerang form, with each end fused with syntergosternite VII + VIII to form single ring of syntergosternite VI + VII + VIII; distal part of syntergosternite VI + VII + VIII bowl-shaped, wider than basal part, more or less flat, with two rounded elongate projections and two pairs of spiracles at articulation with fused sternite VI and sternite VII. Syntergosternite VI + VII + VIII basally connected to sternite V; distal margin extended to epandrium.

*Genitalia* (Figure 2, Figure 3 and Figure 4). Hypandrium (sternite IX) in form of concave trapezoidal plate (Figure 2); lateral arms of hypandrium long, hockey-stick-shaped in frontal view, articulated with subepandrial sclerites and epandrium (tergite IX). Pregonites present, postgonites indistinct (Figure 3); pregonite long and wide in lateral view. Epiphallus, distiphallus and basiphallus absent. Phallapodeme large, wide, semilunar in form. Ejaculatory apodeme absent. Epandrium hemispherical, with large posteromedian notch (Figure 4). Surstylus of medium size, dilated caudally; lateral basal lobe articulated with epandrium, medial basal lobe articulated with subepandrial sclerite; apical lobe stout and massive, pointed at apex (Figure 4). Cerci oval, dilated basally, completely fused medially, with two small apical lobes (Figure 4). Subepandrial sclerite present as two small sclerotized, narrow, concave plates, medially not connected, with small rod-shaped lateral bacilliform sclerite, not forming long process arising from basal part of surstylus (Figure 4).

*Thoracic muscles*. Paired symmetrical conical muscles extend from thorax to lateromedial parts of tergite I + II, and also straight muscles extend from thorax to basal parts of sternite II.

*Abdominal muscles* (Figure 1). ITM 2–ITM 4, ITM 5a, ITM 5b, ISM 2–ISM 5, TSM 1–TSM 5. Flat, very short muscles ITM 2–ITM 4 extend from distal parts of tergites II–IV along entire width to basal margins of tergites III–V. Paired symmetrical muscles ITM 5a extend from laterodistal parts of tergite V to lateral parts of syntergosternite VI + VII + VIII. Paired symmetrical muscles ITM 5b extend from lateral parts of tergite V to distal part of lateral projections of syntergosternite VI + VII + VIII.

Paired symmetrical muscles ISM 2–ITM 4 extend along entire basal margin of sternites II–IV to basal margins of sternites III–V, respectively. Very dense, paired, symmetrical, long muscles ISM 5 extend from basal margins of sclerotized plates of sternite V to laterobasal margin of synsternite VI + VII. Wide and flat pleural abdominal muscles TSM 1–ITM 5 easily discernible on corresponding segments.

*Pregenital muscles* (Figure 1, Figure 2 and Figure 4). M 18, M 19 ^1^, M 19 ^2^.

Paired symmetrical muscles M 18 extending from lateromedial parts of syntergosternite VI + VII + VIII to lateral parts of basal margin of hypandrium.

Paired some asymmetrical muscles M 19 ^1^ extending from inner surface of lateral parts of syntergosternite VI + VII + VIII to lateral margin of epandrium: right muscle M 19 ^1^ weaker than left M 19 ^1^. Muscle M 19 ^2^ dense, fan-shaped, extending from medial part of basal margin of syntergosternite VI + VII + VIII to middle of basal margin of epandrium.

*Genital muscles.* Tergosternal muscles (Figure 2 and Figure 4). M 5. Paired, symmetrical long muscles M 5 extend from lateral parts of lateral arms of hypandrium to laterobasal parts of basal margin of epandrium.

*Muscles of hypandrial complex* (Figure 2 and Figure 3). M 1, M 2 ^1^, M 2 ^2^. Wide and powerful, paired, symmetrical muscles M 1 extend from hypandrium, occupying considerable part of inner surface of medial part, to curve of basal part of phallapodeme. Long, dense, paired symmetrical muscles M 2 ^1^ extend from distal parts of arms of hypandrium to laterodistal parts of phallapodeme. Paired symmetrical muscles M 2 ^2^ long, dense, extending broadly from basal margin of pregonites to medial and distal parts of phallapodeme.

*Muscles of epandrial complex* (Figure 4). M 3, M 4, M 7, M 24–26. Dense, paired, symmetrical muscles M 3 extend from inner surface of laterobasal parts of epandrium to inner surface of bacilliform sclerite. Fan-shaped, paired, symmetrical muscles M 4 extend from lateral parts of inner surface of epandrium to inner surface of basal extensions of surstyli. Paired, symmetrical, elusive cercal muscles M 7 extend from inner part of subepandrial sclerite to laterobasal parts of cerci. Broad dense muscle M 24 passes inside cerci connecting lateral parts of two halves of cerci. Paired broad muscles M 25 extend from mediobasal part of epandrium to integument of anus. Long paired symmetrical muscles M 26 extend from mediobasal parts of epandrium (slightly more lateral than insertion of muscles M 25) to lateral parts of basal margin of cerci.

### 3.2. Fannia Canicularis (Linnaeus, 1761)

#### 3.2.1. Material Examined

Four males, Russia, Kurgan region, Lebyazhye district, Lisje village, 55°08′ N 66°47′ E, 15.viii. 019, leg. V. Sorokina.

#### 3.2.2. Remarks

The sternite V is a large plate, as long as broad, with a small median notch (Figure 5C). The pregonites are large, strongly sclerotized and fused together apically. The postgonites are shoe-shaped, posterior to the pregonites and narrowly closely associated with hypandrium (Figure 5). The ejaculatory apodeme is absent. The subepandrial sclerite without distinct lateral bacilliform sclerite. The muscles of this species and *F. subpellucens* are the same.

### 3.3. Fannia Incisurata (Zetterstedt, 1838)

#### 3.3.1. Material Examined

Two males, Russia, Kurgan region, Lebyazhye district, Lisje village, 55°08′ N 66°47′ E, 15.viii.2019, leg. V. Sorokina.

#### 3.3.2. Remarks

The sternite V is a large plate, a little broader than long, with a narrow and shallow median notch (Figure 6A). The pregonites are large, sclerotized and narrowly fused medially. The postgonites and ejaculatory apodeme are absent. The subepandrial sclerite has a spirally bent lateral bacilliform sclerite, the tip of which does not connect with the hypandrial arms but is directed more or less parallel to the surstyli (Figure 6). The muscles of this species and *F. subpellucens* are the same.

## 4. Discussion

Analysis of our results shows that the Fanniidae have symmetrical pregenital segments, unlike not only other members of the muscoid grade but also the Oestroidea. There is a significant reduction of the sclerites and muscles of the pregenital segments and the male genitalia of the Fanniidae, which appears to be a derived condition in comparison to the majority of Calyptrata. Within the Calyptratae, only the Hippoboscoidea, like the Fanniidae, have significant parallel reduction of the sclerites and muscles of the pregenital segments and the male genitalia and symmetrical pregenital segments.

In particular, the pregenital sclerites of males of *Fannia* Robineau-Desvoidy, 1830 are strongly reduced and fused into one sclerotized and symmetrical ring. This ring includes syntergosternite VI + VII + VIII, muscles ITM 5a and ITM 5b extending from the tergite V and muscles ISM 5 extending from sternite V to this ring, as well as muscles M 19 ^1^ and M 19 ^2^ extending from this ring to the epandrium and muscles M 18 extending to the hypandrium. In the remaining muscoids, as well as in the Calliphoridae [43] and Tachinidae [44], tergite VI, syntergosternite VII + VIII, sternite VI and sternite VII are clearly developed to varying degrees.

The genital skeleton in the studied species of the Fanniidae is similar to that in the Muscidae, Anthomyiidae [35], Calliphoridae [43] and Tachinidae, except for the aedeagal complex. Most genera of these families possess well-developed pregonites, postgonites, phallapodeme, distiphallus, basiphallus, epiphallus and ejaculatory apodeme, while *Fannia* species possess only the phallapodeme, pregonites and at the most reduced postgonites. Although many researchers have stated that pregonites are absent in the Fanniidae, but postgonites are present [1,2,4,6], we believe that the Fanniidae, on the contrary, possess pregonites and reduced postgonites, or the latter are absent. Pregonites were found in the species studied by us, including *F. canicularis*, and their presence was confirmed by the muscles M 2 ^2^ extending from the basal margin of the pregonites to the medial and distal parts of the phallapodeme. The same long, dense, quite wide muscles M 2 ^2^ were found in the Muscidae previously studied by us, in Anthomyiidae [35], and also Tachinidae and Calliphoridae [43], which, unlike Fanniidae, have well-developed postgonites. The muscles extending from the basal margin of the postgonites to the phallapodeme are only found in Anthomyiidae and Calliphoridae, but these muscles are smaller and weaker than the pregonal muscles and attach to the basal parts of the phallapodeme. Besides that, the Glossinidae and the Nycteribiidae of the superfamily Hippoboscoidea possess well-developed pregonites with muscles extending to the phallapodeme, with postgonites absent [45]. According to Schlein and Theodor [45], these sclerites and their associated muscles are homologous to pregonites and relevant muscles of *Calliphora* Robineau-Desvoidy, 1830. Other families of the Hippoboscoidea, in particular the Hippoboscidae and the Streblidae, possess postgonites and pregonites not having muscles extending to the phallapodeme [45]. Thus, pregonites having muscles M 2 ^2^ which extend to the phallapodeme were found in all of the Calyptrata (muscoid grade, Oestroidea and Hippoboscoidea). The presence of pregonites and muscles M 2 ^2^ can be considered as the ground-plan character state of the Calyptratae. Additionally, in most calyptrates that have been studied, the postgonites are smaller in size than the pregonites and are located posterior to them. Small sclerites below and posterior to the pregonites in *F. subpellucens* (Figure 3) and *F. canicularis* (Figure 5) are, in our opinion, reduced postgonites. However, these small sclerites could be the postgonal apodeme which is well developed in most oestroid calyptrates [40]. These statements persuade us to consider the large sclerites of the aedeagal complex to be pregonites.

One more diagnostic element within the Fanniidae is the bacilliform sclerite, which is considered an autapomorphous character in the ground plan of the muscoid grade [24]. According to the revised epandrial hypothesis [40,46], the bacilliform sclerite is part of the subepandrial region, which is often prolonged as a pair of rod-like extensions and articulates with the posterolateral corners of the epandrium or base of the surstyli. In the Fanniidae, the subepandrial region presents a pair of reduced subepandrial sclerites that articulate with the phallus + hypandrium and a pair of lateral bacilliform sclerites associated with the surstyli. The distal parts of bacilliform sclerites may be modified into long processes arising from the base of the surstylus that varies in shape [2] or reduced to small sclerites, sometimes indistinct (e.g., as in *F. canicularis*). In the other muscoids the subepandrial sclerite is developed as a desclerotized fold or as sclerotized plates, but the lateral bacilliform sclerite is reduced and indistinct. However, the bacilliform sclerites (processus longus) is well developed in the Calliphoridae, and its tip articulates with the hypandrial arms and surstyli [40,43]. These data confirm the plesiomorphic condition of the lateral bacilliform sclerite for the Fanniidae and contradicts the claim that the bacilliform sclerite is an autapomorphous character state in the ground plan of the muscoid grade.

In the Fanniidae, the muscle set extending from the pregenital sclerites to the epandrium, and the hypandrium is in many ways similar to that of the Muscidae, Anthomyiidae, Scathophagidae, Tachinidae and Calliphoridae. Differences in the muscles of the Fanniidae and the other Calyptrata that we studied were also found. The muscles M 18 are symmetrical, and the muscles M 19 ^1^ and M 19 ^2^ are almost symmetrical in the Fanniidae, unlike the completely asymmetrical pair in the other families.

The pregenital muscles of the Hippoboscoidea have been studied in insufficient detail [45,47,48] that does not allow a full comparison with those of the Fanniidae. According to the available data, the pregenital muscles of the Fanniidae are most similar with those of the Hippoboscidae, which have a pair of muscles (M 18) extending from the pregenital segments to the hypandrium and two pairs of muscles (M 19) extending to the epandrium. In other families of the Hippoboscoidea the set of pregenital muscles is less. One pair of the tergosternal muscles M 5 connecting the hypandrium and the epandrium has been indicated in all families of the Hippoboscoidea, like the Fanniidae.

The musculature of the epandrium of the Fanniidae is generally similar to that of the Muscidae [36,37,38,39]. The main difference is in the shape and position of the cercal muscles M 26. In *Fannia*, as well as in *Tachina* Meigen, 1803 [44], the muscles M 26 extend from the mediobasal parts of the epandrium to the lateral parts of the basal margin of the cerci (Figure 4A). In the Muscidae, the muscles M 26 are shorter and more fan-shaped; they extend from distolateral parts of the epandrium (distal to muscles M 3 and M 4) to more lateral cercal outgrowths. The muscles M 3 of the subepandrial region and muscles M 4 of the surstyli in the Fanniidae attach to the epandrium, more lateral to the muscles M 26, but in the Muscidae, the attachment is more basal to muscles M 26. In Anthomyiidae and Calliphoridae, the muscles M 26 are located between the more distal muscles M 3 ^2^ and the more basal muscles M 3 ^1^ and M 4. In the Hippoboscoidea, the muscles set of the epandrial complex is less than in the Fanniidae and the other studied Calyptrata [46,47,48].

The Fanniidae have three pairs of muscles M 19 (paired M 19 ^1^ and unpaired M 19 ^2^) as do primitive Muscidae [39], *Tachina* [44] and *Calliphora* [43]. This character state we consider plesiomorphic for the Calyptrata.

The muscle set of the hypandrial complex turned out to be different within members of the muscoid grade. *Delia platura* (Meigen, 1826), according to Hennig [35], has the largest set of phallapodeme muscles (six pairs) from the hypandrium, pregonites, epiphallus and postgonites: M 1, M 2 ^1^, M 2 ^2^ a, M 2 ^2^ b, M 2 ^3^ and M 2 ^4^ (in Hennig: M 41, M 35–37, M 38, M 40, M 39) and the ejaculatory apodeme muscle M 23. *Scathophaga stercoraria* (Linnaeus, 1758), according to Ovtshinnikova [33], has five pairs of phallapodeme muscles: M 1, M 41, M 2 ^1^, M 2 ^2^, M 2 ^3^ (from the hypandrium, pregonites and the epiphallus) and the ejaculatory apodeme muscle M 23. However, *Scathophaga* has the same set of muscles of the hypandrial complex because it includes muscle M 42 extending from the hypandrium to the pregonites, which is absent in *Delia platura*. The phallapodeme muscles M 2 ^2^ a and M 2 ^4^ are absent in *Scathophaga*. The Muscidae have the same set of phallapodeme muscles as *Scathophaga*, but this set varies among the subfamilies. The larger muscle set is in the Azeliinae [39].

Within the Oestroidea that have been studied, the largest set of muscles of the hypandrial complex was found by Salzer [43] in *Calliphora erythrocephala* (Meigen, 1826), and this set was the same as in *Delia platura*. The same set of muscles of the hypandrial complex was also found in *Tachina* [44] except for a pair of postgonite muscles M 2 ^4^, which is absent in this genus.

In the Fanniidae, the set of phallapodeme muscles is less than in other muscoids and in the Oestroidea, and it includes only muscles M 1 and M 2 ^1^ extending from the hypandrium and muscles M 2 ^2^ extending from the pregonites. The greatest similarity of the set of phallapodeme muscles was found between the Fanniidae and the Glossinidae and the Nycteribiidae; all of these have three pairs of phallapodeme muscles (M 1, M 2 ^1^, M 2 ^2^). In the Hippoboscidae and the Streblidae, only two pairs of phallapodeme muscles (M 1, M 2) have been found. Such a set of phallapodeme muscles of the Fanniidae, as well as a fusion of the pregenital sclerites and the absence of parts of the phallus, can be considered to be an apomorphic state. 

In our opinion, the complete set of phallapodeme muscles in the Anthomyiidae represents the basal state (plesiomorphic), and the structure of the genital sclerites and muscles in Fanniidae, as well as in most Muscidae, therefore reveals a certain degree of reduction that represents the advanced state (apomorphic).

## 5. Conclusions

The genital and pregenital modifications that we have detected in the Fanniidae, in particular the symmetry of the pregenital segments and muscles, the reduction of the pregenital and genital sclerites and musculature, including the phallapodeme muscles, thus confirm the derived condition (apomorphic state) of this family. On the other hand, the presence of the lateral bacilliform sclerites, as well as the presence of epandrial muscles M 26, three muscles M 19 (paired M 19 ^1^ and unpaired M 19 ^2^) and paired muscle M 18 as in primitive Muscidae and also in *Tachina* and *Calliphora*, can be considered as a plesiomorphic state. The position of the Fanniidae in the calyptrate phylogeny has been further corroborated by the structure of the sclerites and muscles of the male abdominal segments and terminalia. Our results confirmed the molecular data which places the Fanniidae as the sister group to a clade consisting of the Oestroidea plus remaining muscoids [25,26,27,30,31].

## Figures and Tables

**Figure 1 insects-13-00210-f001:**
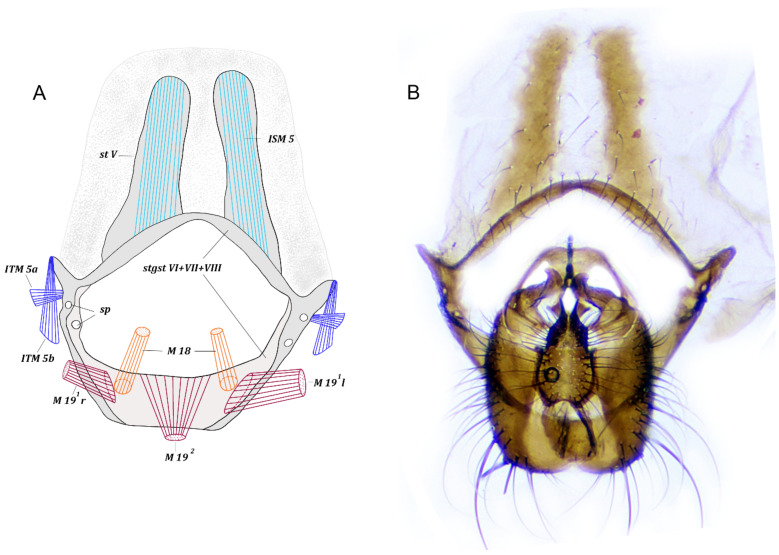
*Fannia subpellucens* (Zetterstedt), male. (**A**) Sternite 5 and pregenital segments, inner view, muscles scheme. (**B**) Sternite 5, pregenital and genital segments, ventral view.

**Figure 2 insects-13-00210-f002:**
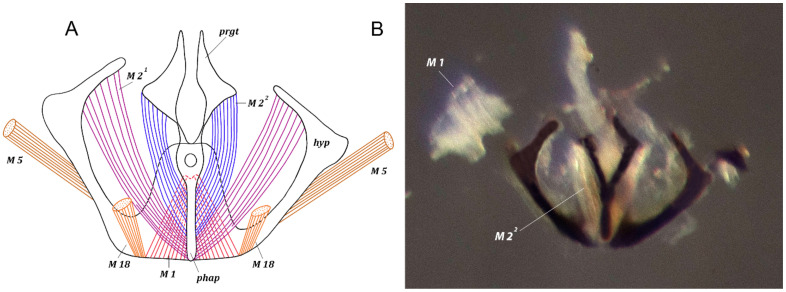
*Fannia subpellucens* (Zetterstedt), male, hypandrial complex, frontal view. (**A**) Muscles scheme. (**B**) Digital image, muscles M 2 ^1^ removed, muscle M 1 displaced.

**Figure 3 insects-13-00210-f003:**
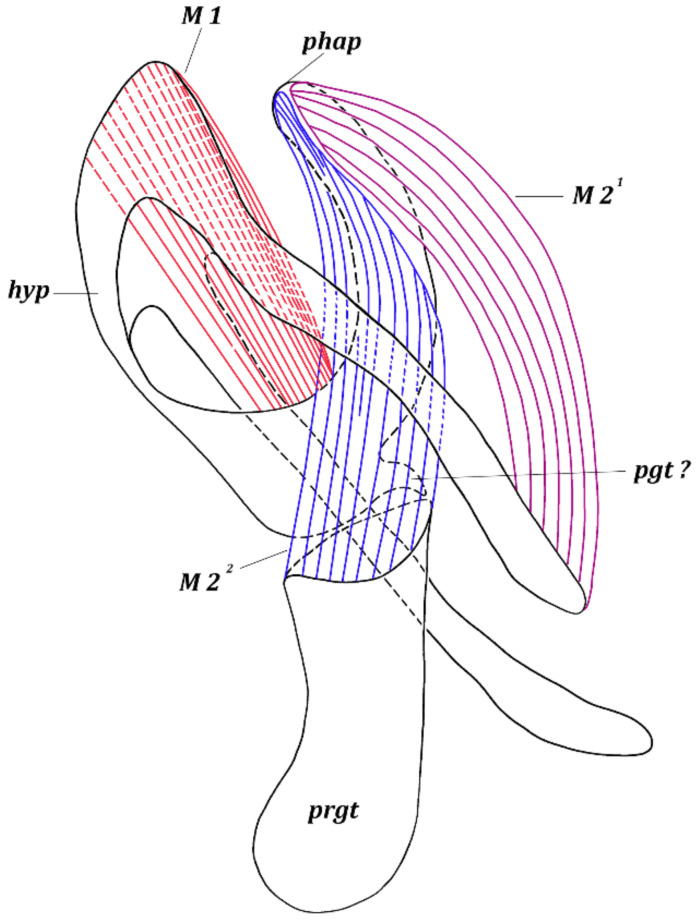
*Fannia subpellucens* (Zetterstedt), male, hypandrial complex, lateral view, muscles scheme.

**Figure 4 insects-13-00210-f004:**
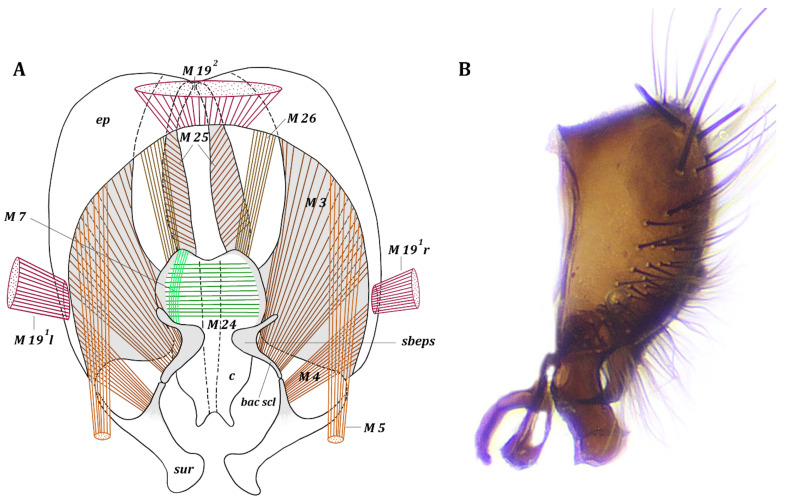
*Fannia subpellucens* (Zetterstedt), male. (**A**) Epandrial complex, inner view, muscles scheme. (**B**) Genitalia, lateral view.

**Figure 5 insects-13-00210-f005:**
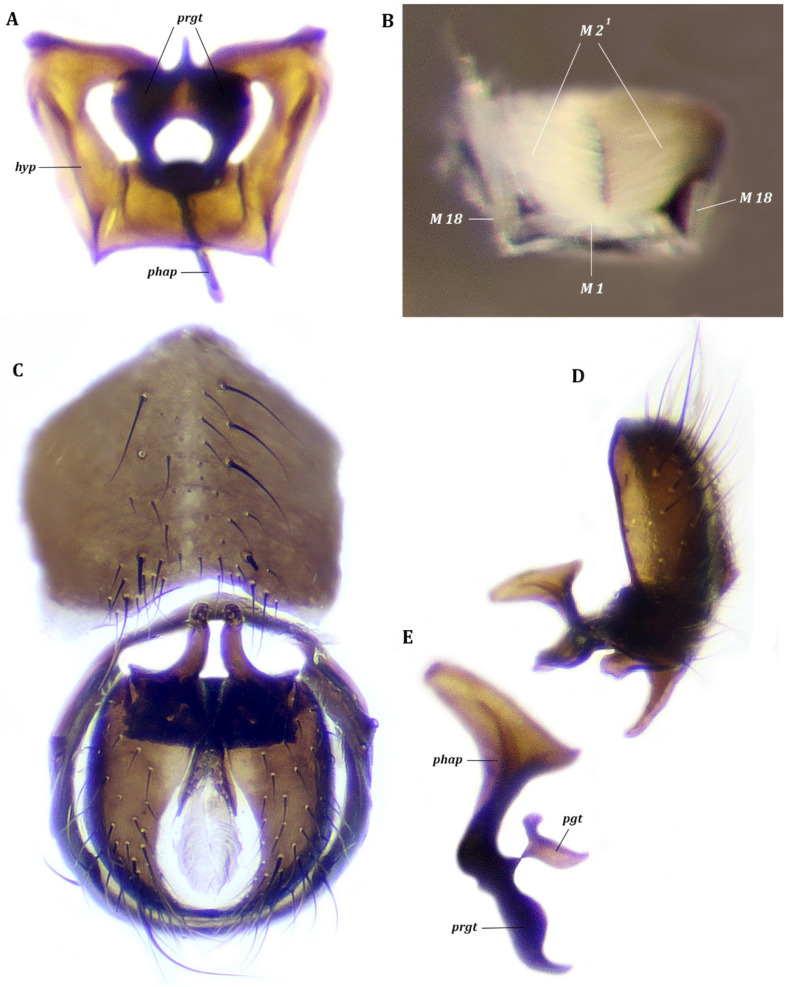
*Fannia canicularis* (Linnaeus), male. (**A**) Hypandrial complex, frontal view. (**B**) Hypandrial complex, frontal view, with muscles M 18, M 1 and M 2 ^1^. (**C**) Epandrial complex, dorsal view. (**D**) Terminalia, lateral view. (**E**) Aedeagal complex, lateral view.

**Figure 6 insects-13-00210-f006:**
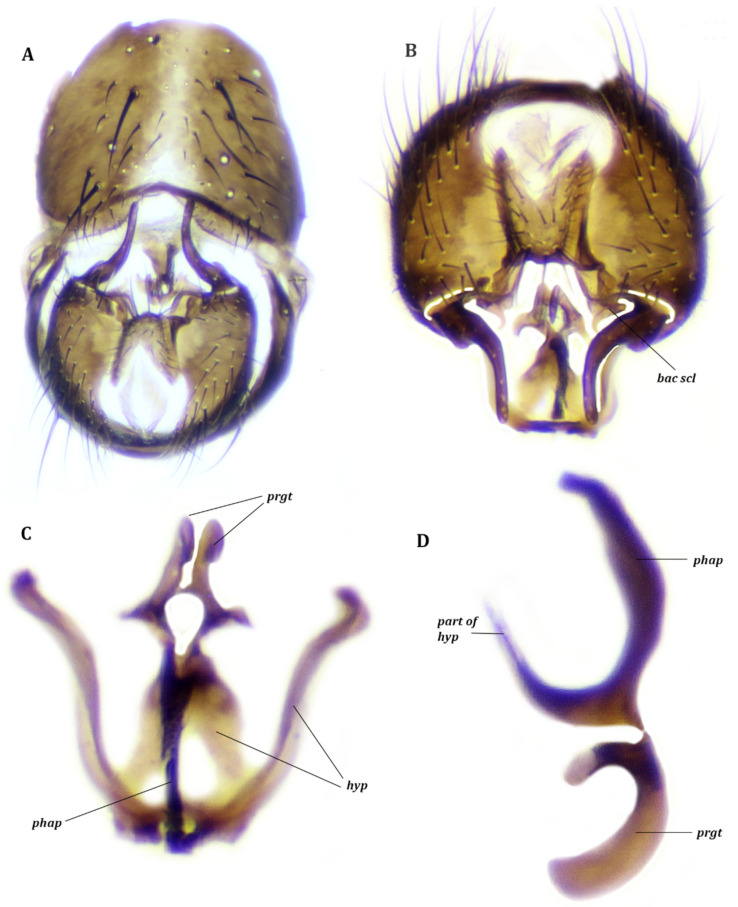
*Fannia incisurata* (Zetterstedt), male. (**A**) pregenital and genital segments, ventral view. (**B**) Terminalia, dorsal view. (**C**) Hypandrial complex, frontal view. (**D**) Aedeagal complex, lateral view.

## Data Availability

Not applicable.

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
