# Peer review of "The Phylogenetic Relationships of the Fanniidae within the Muscoid Grade (Diptera: Calyptrata) Based on the Musculature of the Male Terminalia"

_insects, 2022, doi:10.3390/insects13020210_

Round 1

Reviewer 1 Report

Review of MS: “The phylogenetic relationships of the Fanniidae within the Muscoidea (Diptera) based on the musculature of the male terminalia”.

I have structured this review in a section with more general comments and a section with more specific recommendations on what to do next.

Comments to the MS:

This MS documents the structure of the sclerites and associated muscles of the male terminalia of three species of the family Fanniidae.

This type of base-line data is valuable as part of the ongoing documentation of Diptera morphology, and it is particularly welcome because skeleto-muscular studies are sparse and require a unique expertise that builds on many years of study.

Photographs could be of better resolution and with more depth of focus, and I wonder why figures like 1B and 6A have not been rotated to have their axis changed from near-vertical to vertical? This is mainly cosmetics, but it would take just a few minutes.

I do, however, have some more serious hesitations, because the text suffers from vague and at places even imprecise wording, and the analytical argumentation is often shallow or even misdirected. I am giving most of those cases in the following, but others are annotated directly in the MS.

The MS states for the Fanniidae that “its systematic position within the Calyptrata and the Muscoidea is still unsettled”.

--- This is not quite fair considering two recent molecular analyses (both cited in the MS) with massive datasets, which point very strongly to a position of the Fanniidae at the base of a ‘muscoid grade’, which is paraphyletic with regard to the Oestroidea. This also means that the “Muscoidea” currently is recognized as a paraphyletic ‘non-taxon’, and it is highly recommended to avoid using this as a taxon name. A better term would be ‘muscoid grade’, referring to the assemblage of families Fanniidae, Muscidae, Anthomyiidae and Scathophagidae, which have a subordinate position relative to the Oestroidea.

The structure of male terminalia is “considered to be decisive in the phylogenetic branching of the Cyclorrhapha”.

--- Male terminalia certainly present many phylogenetically informative characters, but it is going too far to claim that these are “decisive”. There are other character complexes in both adult and larval morphology, which also bring what may be considered strong phylogenetic signals at this level, e.g., larval cephaloskeleton, wing venation, antennal structure, position of adult abdominal spiracles, etc.

The morphological details “are compared with other Muscoidea as well as the Oestroidea”.

--- There are some comparisons across these taxa, but there is also a noteworthy absence of comparisons with any member of the superfamily Hippoboscoidea. This is unfortunate because this would be the appropriate out-group comparison to fully evaluate whether a given feature or character state found in the Faniidae is plesiomorphic or apomorphic. If the Fanniidae are sister to the clade (muscoid grade + Oestroidea), then the polarity of two different character states found in these two clades can only be settled by comparisons to the Hippoboscoidea. Therefore, the statement that “Such a dual state of characters of the Fanniidae suggests that they should be placed on a branch at the base of Calyptrata” is deficient by lacking the proper outgroup comparison to the Hippoboscoidea. A “dual state of characters” simply cannot by itself settle a polarity or evolutionary direction.

“Such structure of the sclerites and muscles of the male abdominal segments and terminalia places the Fanniidae beyond the Muscoidea and Oestroidea”.

    And also:

“The structure of the sclerites and muscles of 36 the male abdominal segments and terminalia place the Fanniidae outside the Muscoidae and Oestroidea”.

--- I suppose the word “beyond” in the first of these sentences is used in the meaning of “outside” or “basal to”, but it is still incorrect. Given that the muscoids represent a paraphyletic grade, the Fanniidae may take a basal position and still be part of that paraphyletic assemblage. Furthermore, it is unclear exactly what is meant by “Such structures”. The preceding sentence mentions “presence of the bacilliform sclerite as well as the presence and position of the epandrial muscles of the Fanniidae as in primitive Muscidae and also in Tachina and Calliphora, confirms their basal state (plesiomorphous)”. However, the sharing of plesiomorphous character states will be silent on phylogenetic position. For the same reason, the statement that “plesiomorphous characters of the Fanniidae [...] suggests that the family diverged at the base of the Calyptrata” is inaccurate because “plesiomorphous characters” by themselves do not carry information about cladistic grouping.

The statement that the “reduction of [several structures] indicates the progressiveness of the group” is not very informative. Any group will possess specialisations, and any of these will be evidence of evolution or ‘progression’. Similarly with the suggestion that: “a reduction of the sclerites and muscles of the pregenital and genital segments and male genitalia of the Fanniidae is the result of apomorphous reduction”. A reduction by definition is apomorphic (the prefix “re-“ means ‘again’ or ‘back to’). And also with the statement that “their further development took place along their progressive evolutionary path”. I suppose evolution is always “progressive” at least in the sense that evolution is directional and moves forward (in time).

The following statement is unclear (at least to this reviewer):

“The homologies of the pregenital sclerites in members of different subfamilies of the Muscidae, in particular the nature of tergite VI, sternites VI and VII, syntergosternite VII + VIII, and of the hypandrial appendages, were confirmed by analysis of the muscle connections. A different set of muscles has been detected within the Muscidae at the subfamily level”.

--- Different in which way? And how can the homologies be "confirmed by analysis of the muscle connections", if the Muscidae have a "different set of muscles"?

“This character confirms the plesiomorphous state of the group.”

--- A single character cannot by itself confirm anything. Every character state will be part of a more or less comprehensive analysis, and as such every character may provide its own evidence, which together with evidence from other characters amount to support (or rejection) of a given hypothesis.

Specifically for the section on Conclusions, the following sentence should be omitted as it mainly states that the taxon Fanniidae possesses apomorphic as well as plesiomorphic character states, which is a statement that holds for every taxon on this planet:

“The genital and pregenital modifications that we have detected in the Fanniidae ... thus confirm the progressiveness (apomorphous state) of this family”.

The Conclusion goes on like this: “On the other hand, the presence of the bacilliform process as well as the presence of epandrial muscles M 26, three muscles M 19 (paired M 191 and unpaired M 192) and paired muscle Ðœ 18 as in primitive Muscidae and also in Tachina and Calliphora, confirms their basal state (plesiomorphous). Such a dual state of characters of the Fanniidae suggests that they should be placed on a branch at the base of Calyptrata, and their further development took place along their progressive evolutionary path”.

The presence of bacilliform processes and the same muscular complement as in other calyptrates in the groundplan of Fanniidae only supports that Fanniidae are part of the calyptrates, and certainly does not suggest their specific phylogenetic position within this clade.

Recommendations on what to do next:

There are good skeleto-muscular data in the MS, but I would suggest to change the way this is used as argumentation for the phylogenetic position of Fanniidae.

Today we are seeing phylogenetic analyses based on thousands of genes, which means hundreds of thousands of molecular characters. Molecular phylogenies are getting stronger every year, and while I definitely do not see molecular phylogenies as the only way to get to the ‘true’ phylogeny, we should recognize those that appear to have particularly high support. For the calyptrates, we have studies applying Sanger sequence data as well as comprehensive transcriptome data, and these studies provide strong evidence that the Fanniidae are sister to a clade comprising the other muscoids plus the oestroids (and do not use the taxon name “Muscoidea”, as the muscoid families together form a paraphyletic grade). This molecular phylogeny should be the starting point or ‘null-hypothesis’, and the authors should discuss whether the skeleto-muscular data are in conflict with – or in agreement with – this hypothesis. And as there is evidently full agreement, the authors can conclude that the position of the Fanniidae in the calyptrate phylogeny has been further corroborated.

Decisions on whether character states for the groundplan of Fanniidae are plesiomorphic or apomorphic should not (and cannot) be argued from comparisons only with members of the clase muscoids + oestroids. The proper outgroup comparison would be with members of the Hippoboscoidea, but it could also be with less specialised members of one of the acalyptrate families.

More specific issues:

The claim that the groundplan of the Fanniidae has a pair of well-developed pregonites and reduced or undeveloped postgonites is very interesting, as this goes counter to other studies of the male genitalia in Fanniidae. Therefore, it would be nice to see a more elaborate discussion of the evidence. How do the authors recognize a structure as either a pre- or a postgonite? Apart from the muscular data, would there be any other evidence? Why have all other studies erred on this interpretation?

I would also recommend to take a new look at the bacilliform sclerites. These are currently defined as the lateral parts of the subepandrial sclerite, when the median part of the latter is membranous. Bacilliform sclerites belong to the groundplan of the calyptrates, and they should therefore be expected to be found in the Fanniidae unless they are secondarily reduced. The MS states for Fannia subpellucens: “Subepandrial sclerite present as two small sclerotized, narrow, concave plates, medially not connected, without bacilliform sclerite”. These “narrow, concave plates, medially not connected” are the bacilliform sclerites. In many Fanniidae the bacilliform sclerites have evolved into complicated structures, but in Fannia subpellucens they look like what we see in a large number of calyptrates.

--------------------------------------

Author Response

We thank the referee for comments. We took into account most of the comments and made the appropriate edits or additions to the manuscript. We would like to answer some remarks by explaining our position. You can find it below and also, please see the attachment with responds in the pdf file.

Point 1: Photographs could be of better resolution and with more depth of focus, and I wonder why figures like 1B and 6A have not been rotated to have their axis changed from near-vertical to vertical? This is mainly cosmetics, but it would take just a few minutes.

Response 1: done

Point 2: The MS states for the Fanniidae that “its systematic position within the Calyptrata and the Muscoidea is still unsettled”.

--- This is not quite fair considering two recent molecular analyses (both cited in the MS) with massive datasets, which point very strongly to a position of the Fanniidae at the base of a ‘muscoid grade’, which is paraphyletic with regard to the Oestroidea. This also means that the “Muscoidea” currently is recognized as a paraphyletic ‘non-taxon’, and it is highly recommended to avoid using this as a taxon name. A better term would be ‘muscoid grade’, referring to the assemblage of families Fanniidae, Muscidae, Anthomyiidae and Scathophagidae, which have a subordinate position relative to the Oestroidea.

Response 2: we agree, “unsettled” is not a suitable expression and we changed it. Yes, of course, last molecular analyses point a position of the Fanniidae at the base of a ‘muscoid grade’, we don't mind it. Kutty et al. (2008, 2019) suggested to use ‘muscoid grade’ instead of Muscoidea, but we have not seen that this term widely used in the dipterological community. Beside there is not still new established classification of the Calyptrata, so we prefer to use traditional terms for the time being. The main aim of our work is to study the muscles and to clarify phylogenetic relationships. In our opinion, we have the right to discuss the existing classifications and accept them or not.

Point 3: The structure of male terminalia is “considered to be decisive in the phylogenetic branching of the Cyclorrhapha”.

--- Male terminalia certainly present many phylogenetically informative characters, but it is going too far to claim that these are “decisive”. There are other character complexes in both adult and larval morphology, which also bring what may be considered strong phylogenetic signals at this level, e.g., larval cephaloskeleton, wing venation, antennal structure, position of adult abdominal spiracles, etc.

Response 3: we changed word “decisive” on “one of decisive”. But the male genitalia indeed represent not only a lot of phylogenetically informative characters, but have the important in copulation rotation to 360 degrees, resulting in circumversion of the terminalia. This rotation has given many changes in the structure of the male genitalia and many new opportunities for the evolutionary development of Cyclorrhapha.

Point 4: The morphological details “are compared with other Muscoidea as well as the Oestroidea”.

--- There are some comparisons across these taxa, but there is also a noteworthy absence of comparisons with any member of the superfamily Hippoboscoidea. This is unfortunate because this would be the appropriate out-group comparison to fully evaluate whether a given feature or character state found in the Faniidae is plesiomorphic or apomorphic. If the Fanniidae are sister to the clade (muscoid grade + Oestroidea), then the polarity of two different character states found in these two clades can only be settled by comparisons to the Hippoboscoidea. Therefore, the statement that “Such a dual state of characters of the Fanniidae suggests that they should be placed on a branch at the base of Calyptrata” is deficient by lacking the proper outgroup comparison to the Hippoboscoidea. A “dual state of characters” simply cannot by itself settle a polarity or evolutionary direction.

Response 4: The aim of our manuscript is to study of the sclerites and muscles of the male abdominal segments and terminalia in members of the family Fanniidae and show their relationships within the Muscoidea. A detailed discussion of the systematic position of the Fanniidae within the Calyptrata is currently impossible now and does not the aim of this study, because this requires studying the muscles of other families, including the Hippoboscoidea. It is our plans and we wrote about it in the “Conclusions”. However, we took into account the comment of the reviewer and added in the “Discussion” some information about the Hippoboscoidea from known references. This information confirmed the plesiomorphic or apomorphic state of some characters. For example, the presence of pregonites with muscles extending to the phallapodeme and the presence of well-developed pregenital muscles extending to the epandrium and hypandrium are plesiomorphic state.

The sentence “Such a dual state of characters of the Fanniidae suggests that they could be placed on the base of Calyptrata” is not the statement, it is only our hypothesis which can be expressed in the framework of this work.

Point 5: “Such structure of the sclerites and muscles of the male abdominal segments and terminalia places the Fanniidae beyond the Muscoidea and Oestroidea”.

And also:

“The structure of the sclerites and muscles of 36 the male abdominal segments and terminalia place the Fanniidae outside the Muscoidae and Oestroidea”.

--- I suppose the word “beyond” in the first of these sentences is used in the meaning of “outside” or “basal to”, but it is still incorrect. Given that the muscoids represent a paraphyletic grade, the Fanniidae may take a basal position and still be part of that paraphyletic assemblage. Furthermore, it is unclear exactly what is meant by “Such structures”. The preceding sentence mentions “presence of the bacilliform sclerite as well as the presence and position of the epandrial muscles of the Fanniidae as in primitive Muscidae and also in Tachina and Calliphora, confirms their basal state (plesiomorphous)”. However, the sharing of plesiomorphous character states will be silent on phylogenetic position. For the same reason, the statement that “plesiomorphous characters of the Fanniidae [...] suggests that the family diverged at the base of the Calyptrata” is inaccurate because “plesiomorphous characters” by themselves do not carry information about cladistic grouping.

Response 5:  Our work is a morphological study, not cladistic study, so this comment, on our opinion, is debatable. The aim of this work is to study the muscles of the Fanniidae and compare them with related families. We did not build cladograms and did not use corresponded terms, example, like “sister group”. We named characters as plesiomorphic or apomorphic based on the study and comparison of muscles of different Calyptrata. And we only put forward the hypothesis about the position of the Fanniidae relative to other muscoids. This hypothesis can be further criticized or accepted by any reader.

Point 6: The statement that the “reduction of [several structures] indicates the progressiveness of the group” is not very informative. Any group will possess specialisations, and any of these will be evidence of evolution or ‘progression’. Similarly with the suggestion that: “a reduction of the sclerites and muscles of the pregenital and genital segments and male genitalia of the Fanniidae is the result of apomorphous reduction”. A reduction by definition is apomorphic (the prefix “re-“ means ‘again’ or ‘back to’). And also with the statement that “their further development took place along their progressive evolutionary path”. I suppose evolution is always “progressive” at least in the sense that evolution is directional and moves forward (in time).

Response 6:  we changed these expressions.

Point 7: The following statement is unclear (at least to this reviewer):

“The homologies of the pregenital sclerites in members of different subfamilies of the Muscidae, in particular the nature of tergite VI, sternites VI and VII, syntergosternite VII + VIII, and of the hypandrial appendages, were confirmed by analysis of the muscle connections. A different set of muscles has been detected within the Muscidae at the subfamily level”.

--- Different in which way? And how can the homologies be "confirmed by analysis of the muscle connections", if the Muscidae have a "different set of muscles"?

Response 7: The homologies and a "different set of muscles" are different things in this context they are not connected in any way.

Among the morphological characters used in phylogenetic reconstructions and classification systems, the characters describing the morphology of muscles of the genital and pregenital structures are more stable than those of the sclerites (See: Matsuda, 1976; Ovtshinnikova, 1989; Friedrich and Beutel, 2008). Besides, the study of muscles helps to clarify homology and to reveal parallelisms in the pregenital and genital sclerites (See: Ovtshinnikova, 1989, 1994, 2000; Ovtshinnikova, Yeates, 1998; Galinskaya, Ovtshinnikova, 2014, 2015; Ovtshinnikova, Galinskaya, 2016a, 2016b, 2017; Galinskaya et al., 2018; Ovtshinnikova et al. 2018, 2019; Sorokina, Ovtshinnikova, 2020; Ovtshinnikova, Sorokina, 2020, 2021).

For example: Cyclorrhapha is characterized by genital rotation that determines asymmetry in both sclerites and muscles of abdominal segments VI–VIII and partly segment IX. Since the pregenital sclerites of segments VI–VIII are partly reduced, modified, and fused, in our previous studies we used the characteristic features of the musculature to clarify the homologies of some male pregenital sclerites in the family Muscidae. In particular, it was shown that the muscles in Brachycera Orthorrhapha and Acalyptratae usually extended between the consecutive segments: from segment V to VI, from segment VI to VII, from VII to VIII, and from VIII to IX. The nature of tergite VI, sternites VI and VII, and syntergosternite VII + VIII in Muscidae was confirmed by analysis of muscles of the male terminalia.

“A different set of muscles within the Muscidae at the subfamily level” means different muscles numbers of certain structures. For example: the set of the phallapodeme muscles in the Azeliinae is four (M 1, M 21, M 22, M 23), in the Muscinae is two (M 21, M 22), in the Mydaeinae also two but different (M 22, M 23). See: Ovtshinnikova et al., 2018, 2019; Sorokina, Ovtshinnikova, 2020; Ovtshinnikova, Sorokina, 2020.

Point 8: “This character confirms the plesiomorphous state of the group.”

--- A single character cannot by itself confirm anything. Every character state will be part of a more or less comprehensive analysis, and as such every character may provide its own evidence, which together with evidence from other characters amount to support (or rejection) of a given hypothesis.

Response 8: we changed this expression.

Point 9: Specifically for the section on Conclusions, the following sentence should be omitted as it mainly states that the taxon Fanniidae possesses apomorphic as well as plesiomorphic character states, which is a statement that holds for every taxon on this planet:

“The genital and pregenital modifications that we have detected in the Fanniidae ... thus confirm the progressiveness (apomorphous state) of this family”.

Response 9: we changed these expressions.

Point 10: The Conclusion goes on like this: “On the other hand, the presence of the bacilliform process as well as the presence of epandrial muscles M 26, three muscles M 19 (paired M 191 and unpaired M 192) and paired muscle Ðœ 18 as in primitive Muscidae and also in Tachina and Calliphora, confirms their basal state (plesiomorphous). Such a dual state of characters of the Fanniidae suggests that they should be placed on a branch at the base of Calyptrata, and their further development took place along their progressive evolutionary path”.

The presence of bacilliform processes and the same muscular complement as in other calyptrates in the groundplan of Fanniidae only supports that Fanniidae are part of the calyptrates, and certainly does not suggest their specific phylogenetic position within this clade.

Response 10: Our work is a morphological study and we only put forward the hypothesis about the position of the Fanniidae relative to other muscoids according studied muscles. It is not cladistics work and we did not build the cladogram with the position of the Fanniidae. We just discuss about it.

Point 11: There are good skeleto-muscular data in the MS, but I would suggest to change the way this is used as argumentation for the phylogenetic position of Fanniidae.

Response 11: In this MS we do not affirm the phylogenetic position of the Fanniidae but only discuss the apomorphic and plesiomorphic state of the characters and suggest the possibility of their use for phylogenetic reconstructions. We changed some expressions according reviewer’s comments.

Point 12: Today we are seeing phylogenetic analyses based on thousands of genes, which means hundreds of thousands of molecular characters. Molecular phylogenies are getting stronger every year, and while I definitely do not see molecular phylogenies as the only way to get to the ‘true’ phylogeny, we should recognize those that appear to have particularly high support. For the calyptrates, we have studies applying Sanger sequence data as well as comprehensive transcriptome data, and these studies provide strong evidence that the Fanniidae are sister to a clade comprising the other muscoids plus the oestroids (and do not use the taxon name “Muscoidea”, as the muscoid families together form a paraphyletic grade). This molecular phylogeny should be the starting point or ‘null-hypothesis’, and the authors should discuss whether the skeleto-muscular data are in conflict with – or in agreement with – this hypothesis. And as there is evidently full agreement, the authors can conclude that the position of the Fanniidae in the calyptrate phylogeny has been further corroborated.

Response 12: We do not deny this, so it would have been written in the Сonclusion. Since the reviewer did not understand this, we changed the expressions.

Point 13: Decisions on whether character states for the groundplan of Fanniidae are plesiomorphic or apomorphic should not (and cannot) be argued from comparisons only with members of the clase muscoids + oestroids. The proper outgroup comparison would be with members of the Hippoboscoidea, but it could also be with less specialised members of one of the acalyptrate families.

Response 13: The aim of our manuscript is to study of the sclerites and muscles of the male abdominal segments and terminalia in members of the family Fanniidae and show their relationships within the Muscoidea. A detailed discussion of the systematic position of the Fanniidae within the Calyptrata is currently impossible now and does not the aim of this study, because this requires studying the muscles of other families, including the Hippoboscoidea. It is our plans and we wrote about it in the “Conclusions”. However, we have added in the “Discussion” some information about the Hippoboscoidea from known references. This information confirmed the plesiomorphic or apomorphic state of some characters. For example, the presence of pregonites with muscles extending to the phallapodeme and the presence of well-developed pregenital muscles extending to the epandrium and hypandrium are plesiomorphic state.

Point 14: The claim that the groundplan of the Fanniidae has a pair of well-developed pregonites and reduced or undeveloped postgonites is very interesting, as this goes counter to other studies of the male genitalia in Fanniidae. Therefore, it would be nice to see a more elaborate discussion of the evidence. How do the authors recognize a structure as either a pre- or a postgonite? Apart from the muscular data, would there be any other evidence? Why have all other studies erred on this interpretation?

Response 14: As evidence for the presence of pregonites in the MS, in addition to muscles, the size and position of these sclerites relative to each other were also discussed. We also supplemented this evidence with information about the Hippoboscoidea.

In our opinion, previous researchers used the traditional name of sclerites within a particular family, which was not confirmed by special studies of the homology of structures. No evidence of the nature of this structure was given in previous publications.

Point 15: I would also recommend to take a new look at the bacilliform sclerites. These are currently defined as the lateral parts of the subepandrial sclerite, when the median part of the latter is membranous. Bacilliform sclerites belong to the groundplan of the calyptrates, and they should therefore be expected to be found in the Fanniidae unless they are secondarily reduced. The MS states for Fannia subpellucens: “Subepandrial sclerite present as two small sclerotized, narrow, concave plates, medially not connected, without bacilliform sclerite”. These “narrow, concave plates, medially not connected” are the bacilliform sclerites. In many Fanniidae the bacilliform sclerites have evolved into complicated structures, but in Fannia subpellucens they look like what we see in a large number of calyptrates.

Response 15: we changed this section in MS.

Reviewer 2 Report

This manuscript continues a series of studies on the musculature of the Calyptrata. They are carefully studied and the illustrations continue to improve with each new study. Each study also has a strong phylogenetic focus and this continues. I have made comments/ corrections directly on the ms and have the additional points:

  1. I modified the descriptive sections to a telegraphic or descriptive style.
  2. I also made several suggestions of better words or slight changes in sentence construction.
  3. Sternite V differs among the species studied, but this was not discussed. This should be corrected.
  4. My major critique concerns the discussion around the subepandrial sclerite and the bacilliform sclerites. In the groundplan of the Eremoneura, a deeply invaginated pouch separates the epandrium and proctiger from the hypandrium and the roof of this space is completely sclerotized and termed subepandrial sclerite – from the posterior corners of the epandrium and articulates with the anterodorsal surface of the phallus. Often the posterolateral corners of the epandrium are prolonged and the subepandrial sclerite also is prolonged as a pair of rod-like extensions, termed the bacilliform sclerites (see schematic in Cumming et al. 1995, fig. 3 (revised epandrial hypothesis). In other words, both the subepandrial sclerite and bacilliform sclerites are homologous structures, but are reduced/ developed among taxa. The two sclerites are readily identified among the Empidoidea (see figs in above reference), but among the higher Cyclorrhapha the sclerites are reduced to primarily a pair of slender sclerites and simply termed bacilliform sclerites. See also Sinclair (2000, fig. 40) – the figure shows the ventral region of the epandrium and subepandrial region. The membrane is sclerotized and divided medially. The anterior region (directed to top of page) represents the subepandrial sclerite (unfortunately not labelled) and the pair of posterolateral narrowed plates are the bacilliform sclerites (as labelled).

            Consequently, I do not agree with the authors interpretation of the reduction or loss of the bacilliform sclerites in Fanniidae. The entire subepandrial region is simply reduced to a pair of rods as found in most Cyclorrhapha. The anterior section of the rods which articulates with the phallus+hypandrium can still be considered the reduced subepandrial sclerite and the apical rods associated with the surstyli are bacilliform sclerites. The sections discussing these sclerites needs to be edited.

Author Response

We are very grateful to the referee for the evaluation of our work and all the corrections. We tried to take into account all the comments and made the appropriate corrections. Special thanks for the detailed clarifications regarding bacilliform sclerites, we have revised these sections.

Point 1: I modified the descriptive sections to a telegraphic or descriptive style.

Response 1: done

Point 2: I also made several suggestions of better words or slight changes in sentence construction.

Response 2: done

Point 3: Sternite V differs among the species studied, but this was not discussed. This should be corrected.

Response 3: We added missed figure and short descriptions of sternite V in “Remarks” to all studied species. But we do not consider it expedient to discuss this in detail in this paper, because sternite V is not used as a diagnostic character for Fanniidae.

Point 4: Consequently, I do not agree with the authors interpretation of the reduction or loss of the bacilliform sclerites in Fanniidae. The entire subepandrial region is simply reduced to a pair of rods as found in most Cyclorrhapha. The anterior section of the rods which articulates with the phallus+hypandrium can still be considered the reduced subepandrial sclerite and the apical rods associated with the surstyli are bacilliform sclerites. The sections discussing these sclerites needs to be edited.

Response 4: We changed this section.

Round 2

Reviewer 1 Report

A number of suggestions are indicated directly in the MS.

The authors should either abandon the taxonomic term Muscoidea completely, or argue for why they want to keep this term, when all modern analyses – even the present one – bring evidence that there is no support for a monophyletic Muscoidea. A paraphyletic assemblage of families can be referred to as the “muscoid grade” or the “muscoid families” or even simply as the “muscoids”, but using the formal term “Muscoidea” is misleading and should be avoided. I have made suggestions for relevant changes in most of the Introduction, but this should be done throughout.

I should like to bring one paragraph in focus:

257                                           There is a significant re-

258   duction of the sclerites and muscles of the pregenital segments of the male genitalia of the

259   Fanniidae, which appears to be a derived condition in comparison to the majority of Ca-

260  lyptrata. Within the Calyptratae only the Hyppoboscoidea, like the Fanniidae, have the

261  similar symmetrical pregenital segments and significant reduction of the sclerites and

262  muscles of the pregenital segments of the male genitalia.

The authors state that the reduction of sclerites and muscles in the Fanniidae is a derived condition, but then make the statement that also the Hippoboscoidea have this type of symmetrical pregenital segments and a significant reduction of the sclerites and muscles. The authors should clearly state that they consider these reductions to have taken place independently in the Hippoboscoidea and the Fanniidae.

I strongly recommend to delete the following sentences, which bring no real arguments and even contain erroneous information:

32                                                                                                                                         Such a dichotomy of

33   characters of the Fanniidae suggests that the family diverged at the base of the Calyptrata and that

34   their further development took place along their derived and independent evolutionary path.

And:

388          The data obtained showed that the Fanniidae have characters in the ground-plan of

389    the Calyptrata. Such a dual state of characters of the Fanniidae suggests that they could

390    be placed on the base of Calyptrata, and their further development took place along their

391    derived and independent evolutionary path.

All taxa will have a mixture of ground-plan (plesiomorphic) and derived (apomorphic) character states. That will by itself no argue for a position of the Fanniidae at the base of Calyptrata – and note that it is not the Fanniidae but the Hippoboscoidea that branch off at the base of the Calyptrata.

Statements like the final one in the conclusion are empty and should also be deleted:

396                    “In order to strengthen the phylogenetic position of the Fanniidae within the

397   Calyptrata, it will be necessary to continue the study of genital segments and muscles in

398   representatives of the remaining calyptrate families.”

That sentence only states that continued studies will bring more data and bring science forward. This can be stated for every single study we make.

Author Response

Thank you very much for your suggestions and the correction of English words. We accepted almost all of them, with the exception of a correction in the descriptions sections, since this section was previously edited by at least two native speakers and they recommended using a telegraphic or descriptive style (without articles).

Point 1: The authors should either abandon the taxonomic term Muscoidea completely, or argue for why they want to keep this term, when all modern analyses – even the present one – bring evidence that there is no support for a monophyletic Muscoidea. A paraphyletic assemblage of families can be referred to as the “muscoid grade” or the “muscoid families” or even simply as the “muscoids”, but using the formal term “Muscoidea” is misleading and should be avoided. I have made suggestions for relevant changes in most of the Introduction, but this should be done throughout.

Response 1: we changed this term.

Point 2: I should like to bring one paragraph in focus:

There is a significant reduction of the sclerites and muscles of the pregenital segments of the male genitalia of the Fanniidae, which appears to be a derived condition in comparison to the majority of Calyptrata. Within the Calyptratae only the Hyppoboscoidea, like the Fanniidae, have the similar symmetrical pregenital segments and significant reduction of the sclerites and muscles of the pregenital segments of the male genitalia.

The authors state that the reduction of sclerites and muscles in the Fanniidae is a derived condition, but then make the statement that also the Hippoboscoidea have this type of symmetrical pregenital segments and a significant reduction of the sclerites and muscles. The authors should clearly state that they consider these reductions to have taken place independently in the Hippoboscoidea and the Fanniidae.

Response 2: We have corrected this statement.

Point 3: I strongly recommend to delete the following sentences, which bring no real arguments and even contain erroneous information:

Such a dichotomy of characters of the Fanniidae suggests that the family diverged at the base of the Calyptrata and that their further development took place along their derived and independent evolutionary path.

And:

The data obtained showed that the Fanniidae have characters in the ground-plan of the Calyptrata. Such a dual state of characters of the Fanniidae suggests that they could be placed on the base of Calyptrata, and their further development took place along their derived and independent evolutionary path.

All taxa will have a mixture of ground-plan (plesiomorphic) and derived (apomorphic) character states. That will by itself no argue for a position of the Fanniidae at the base of Calyptrata – and note that it is not the Fanniidae but the Hippoboscoidea that branch off at the base of the Calyptrata.

Response 3: these sentences were deleted.

Point 4: Statements like the final one in the conclusion are empty and should also be deleted:

In order to strengthen the phylogenetic position of the Fanniidae within the Calyptrata, it will be necessary to continue the study of genital segments and muscles in representatives of the remaining calyptrate families.”

That sentence only states that continued studies will bring more data and bring science forward. This can be stated for every single study we make.

Response 4: the sentence was deleted.